# Chemopreventive Effects of Oral Pterostilbene in Multistage Carcinogenesis of Skin Squamous Cell Carcinoma Mouse Model Induced by DMBA/TPA

**DOI:** 10.3390/biomedicines10112743

**Published:** 2022-10-28

**Authors:** Omchit Surien, Siti Fathiah Masre, Dayang Fredalina Basri, Ahmad Rohi Ghazali

**Affiliations:** 1Center for Toxicology and Health Risk Studies (CORE), Faculty of Health Sciences, Universiti Kebangsaan Malaysia (UKM), Kuala Lumpur 50300, Malaysia; 2Centre for Diagnostic, Therapeutic & Investigative Studies (CODTIS), Faculty of Health Sciences, Universiti Kebangsaan Malaysia (UKM), Kuala Lumpur 50300, Malaysia

**Keywords:** pterostilbene, initiation, promotion, squamous cell carcinoma, skin cancer

## Abstract

Skin squamous cell carcinoma (SCC) is a type of non-melanoma skin cancer. Pterostilbene is a natural compound proven to exhibit various pharmacological properties, including chemo-preventive effects. This study aimed to explore the chemo-preventive effect of oral pterostilbene during initiation, promotion or continuous on multistage skin SCC mouse models induced by 7,12-Dimethylbenz(a)anthracene (DMBA)/12-O-Tetradecanoylphorbol-13-acetate (TPA). The experimental design consists of five groups of female Institute of Cancer Research (ICR) mice, with two control groups of vehicle and cancer. Three oral pterostilbene groups consisted of orally administered pterostilbene during initiation, promotion, or continuously. Oral pterostilbene significantly reduced the number and volume of tumours. Oral pterostilbene demonstrated less severe skin histology changes compared to the cancer control group, with less pleomorphic in the cells and nuclei, and the basement membrane remained intact. Our results showed fewer invasive tumours in oral PT-treated groups than in cancer groups that displayed mitotic bodies, highly pleomorphic cells and nuclei, and basement membrane invasion. The cell proliferation marker (Ki-67) was reduced in oral pterostilbene-treated groups. Overall, oral pterostilbene is a promising chemo-preventive intervention due to its anti-initiation and anti-promotion on skin carcinogenesis. Thus, the potential molecular mechanisms of oral pterostilbene chemo-prevention agent should be explored.

## 1. Introduction

The skin is the human body’s largest organ, and its primary function is to serve as a protective barrier from harmful external responses, including physical and immunological functions [1]. Skin cancer is one of the most frequently diagnosed human skin diseases, with the incidence rate of skin cancer continuously increasing, globally [2]. Skin cancer is generally classified into two major groups: melanoma skin cancer (MSC) and non-melanoma skin cancers (NMSCs), also known as keratinocyte carcinoma. Skin cancer categorised as NMSCs includes basal cell carcinoma (BCC), apocrine carcinoma, Merkel cell carcinoma, squamous cell carcinoma (SCC), sebaceous carcinoma, Kaposi sarcoma, and other rare types of skin cancers [3,4]. However, the majority of NSMCs cases are skin BCC and SCC, which can develop at any age, and on various parts of the body [5]. Skin SCC is more aggressive than BCC; skin SCC has a higher mortality rate and contributes to the most death among NMSCs patients. Skin SCC has a higher tendency for metastasis with more severe complications [6,7]. A poor prognosis has been reported among skin SCC patients with metastasis, with only a 10–20% survival rate over 10 years [8].

According to numerous epidemiological and pre-clinical studies, consuming natural food such as vegetables, fruits, herbs, and spices has been linked to lowering the risk of developing cancer [9,10]. Stilbene-based natural compounds, such as resveratrol and pterostilbene, are widely studied polyphenols with numerous pharmacological properties, including anti-inflammatory, antioxidant, anti-cancer, anti-diabetic, and others [11,12,13,14,15]. However, pterostilbene possesses better pharmacokinetic properties compared to resveratrol, with a longer half-life in the body and greater bioavailability [16,17]. The diversity in the chemical structure between pterostilbene and resveratrol is responsible for pterostilbene’s preferable pharmacokinetic properties, as pterostilbene’s phenyl ring consists of two methoxy groups and one hydroxy group, whereas resveratrol’s chemical structure consists of all three hydroxyl groups attached to the phenyl rings. The existence of these two methoxy groups makes pterostilbene more lipophilic and enhances the hydrophobicity that leads to the increase in cellular uptake [18,19]. Figure 1 shows the chemical structure of pterostilbene and resveratrol. The existence of the two methoxy groups in pterostilbene contributes to the higher lipophilicity compared to resveratrol, as Hansch’s π constant of methoxy groups is −0.02, which is higher than the value of the hydroxy group, which is −0.67. The Hansch’s π constant is associated with lipophilicity, as its value is derived from the partition coefficient, which calculates each substituent group’s individual participation to the partition coefficient of a parent molecule [20,21,22]. In addition, the presence of methoxy groups slows down its metabolism compared to the presence of a hydroxy group with more rapid metabolism, as the compound with methoxy group needs O-demethylation to form detoxification metabolite for the excretion [23,24]. In addition to the favourable pharmacokinetic characteristics of pterostilbene, it has also been shown to exhibit chemo-preventive effects against the carcinogenesis of various cancers including skin, colon, and lung cancer [25,26,27,28].

Carcinogenesis, or the development of cancer, is multistage carcinogenesis that generally consists of initiation, promotion, progression, and lastly, metastasis, which involves genetic mutations and cellular hyperproliferation to form benign neoplastic cells and undergo the transformation into malignant cells that can spread from the primary site to other parts of the body [30]. Therefore, cancer is a disease characterised by abnormal cell growth or cell division that has the tendency to metastasis to other organs or parts of the body via blood or lymphatic vessels [31]. The expression of Ki-67 acts as a proliferation marker that is strongly linked with cell division, growth, and the aggressiveness of the tumour, and overexpression of Ki-67 is associated with poor health conditions in cancer patients [32,33]. The 7,12-dimethylbenz[a]anthracene (DMBA)/12-O-tetradecanoylphorbol-13-acetate (TPA) causing the development of squamous cell carcinoma of the skin in mice is also known as the two-stage skin carcinogenesis model, which is suitable for chemo-preventive study and has been widely used to investigate the effectiveness of skin cancer prevention approaches. The established and well-studied in vivo skin cancer mouse model involves the initiation stage, through exposure to DMBA, which is a highly mutagenic agent, and repeated application of TPAs as a promoter agent during the promotion stage until the formation of the tumour [34,35]. Hence, this skin cancer mouse model protocol allowed the initiation and promotion stages to be distinguished practically, and will enable the chemo-preventive agent’s response, or effect on a specific stage in carcinogenesis of skin squamous cell carcinoma to be accessed. Moreover, the model is highly reproducible, as the tumour formation can be monitored throughout the experimental period without animal sacrifice [34,36]. In this study, we investigated the chemo-preventive effect of oral pterostilbene on specific stages during the initiation, promotion or continuation in multistage carcinogenesis of DMBA/TPA-induced skin squamous cell carcinoma mouse model.

## 2. Materials and Methods

### 2.1. Chemicals

The 7,12-dimethylbenz[*a*]anthracene (DMBA) ≥ 95% and 12-O-tetradecanoylphorbol 13-acetate (TPA) ≥99% (TLC), film or powder were purchased from Sigma Aldrich, Saint Louis, Missouri (MO), USA. Pterostilbene, 98% was purchased from J&K Scientific, Beijing, China.

### 2.2. Animal Experimental Research Design

Our study used female mice of the Institute Cancer Research (ICR) strain, all six- to seven-weeks-old, with a body weight ranging from 28 to 30 g. All the mice were held in plastic cages with tissue paper bedding, with the temperature of the animal house at 24 ± °C, with 12 h of light and 12 h of dark. A total of N = 30 mice were randomly separated into five groups, with each group consisting of six mice (n = 6). A rectangular-shaped area of 3 × 5 cm was shaved on the dorsal skin of each mouse using an electronic shaver, followed by a manual shave by razor, 48 h before the experimental procedure. The first group was the vehicle (Vehicle) control group, which received corn oil via oral gavage as pterostilbene solvent and 70% of acetone applied topically on the dorsal rectangular shaved skin, twice a week, as a solvent for DMBA and TPA. Group 2 served as a cancer control group (Cancer), they received corn oil orally from week 0 to week 24, and 200 nmol/100 µL of DMBA, twice a week for two weeks, from week 2 to 3, followed by 20 nmol/100 µL of TPA from week 4 to week 24, also twice a week. Groups 3, 4, and 5 were the pterostilbene treatment groups: Group 3 (Initiation) was pterostilbene-treated during the initiation by DMBA; Group 4 (Promotion) had the pterostilbene treatment, during the promotion that started after DMBA, or during the first exposure to TPA until the end of the experimental animal period. Group 5 had the continuous treatment of pterostilbene, throughout the exposure to the DMBA and TPA treatments. Pterostilbene with a 50 mg/kg dose of body weight was administered via the oral gavage 30 min before each DMBA or TPA exposure. Figure 2 shows the experimental design for each group, to explore the chemo-preventive effect of oral pterostilbene during the specific stage of carcinogenesis in DMBA/TPA-induced skin squamous cell carcinoma. The body weight and the number of tumours (at least 1mm in diameter) were recorded every week. The diameter of the tumours was measured using a ruler, every week. The tumour volume was counted using the formula 4/3πradius3 with radius = 1/2 diameter. The mice were sacrificed after 24 weeks of the experimental animal period, after the intraperitoneal injection (i.p) of ketamine/xylazine cocktail (KTX), and followed by cervical dislocation. All the animal experimental procedures were approved by the Universiti Kebangsaan Malaysia Animal Ethical Committee (UKMAEC) with the approval number FSK/2019/AHMAD ROHI/27-NOV./1066-NOV.-2019-DEC.-2021.

### 2.3. Hematoxylin and Eosin (H&E Staining)

The samples of skin tissues were washed in cold phosphate-buffered saline (PBS), after their removal, and kept in the 10% buffered formalin at 4 °C for fixation. After 24 h of formalin fixation, the skin tissues were moved to 70% of ethanol at 4 °C, until the next step. Next, the skin tissues were processed through dehydration, fixing, and clearing, by an automated tissue processor. After the skin samples were processed with an automatic tissue processor, they were embedded using paraffin wax to produce tissue blocks. The paraffin-embedded skin tissues were sectioned at 4 µm thickness and paraffin ribbons were floated in a water bath with a temperature of 40–50 °C. The paraffin ribbons were then placed on glass slides and dried in an oven (37 °C) for 15–20 min, and continuedair-dried for approximately 24 h at room temperature. After that, the slides were put through the procedure of hematoxylin and eosin (H&E) staining and mounted with coverslips using dibutylphthalate polystyrene xylene (DPX).

### 2.4. Measurement of Epidermal Thickness

Based on H&E staining images obtained from the light microscope, the thickness of the epidermis layer was measured using the Image J software (Image J version 1.46r; National Institute of Health, Bethesda, Maryland(MD), USA. Epidermal thickness was measured at a minimum of five areas-per-tissue sectioning of H&E staining images to obtain the mean epidermis thickness for each mouse [37]. Then, the mean epidermis thickness for each group was obtained once the mice from all five groups were measured (n = 6).

### 2.5. Histopathological Scoring

Histopathological scoring was conducted using the images of H&E staining of the skin tissues, observed under the light microscope. The skin squamous cell carcinoma (SCC) development process in the DMBA/TPA-induced skin carcinogenesis mouse model involved multistage carcinogenesis, which began with changes to the normal epidermis layer into hyperplasia. This epidermis hyperplasia further developed into papilloma. The papilloma eventually progressed to the SCC. Hence, the score 0 to 3 was given, according to the progression in multistage carcinogenesis of skin SCC development in this model, with normal epidermis scored at 0; epidermal hyperplasia: 1, papilloma: 2, and the SCC: 3 [34,38]. The histopathological scoring was conducted with a minimum of five areas or skin lesions, under the 40X magnification of H&E staining images, from each mouse, in order to obtain the mean histopathological scoring for that mouse. Then, the mean histopathological scoring for each group was obtained for all five groups (n = 6).

### 2.6. Immunohistochemistry Staining of Ki-67

Immunohistochemistry staining was conducted using the paraffin-embedded skin tissue sections, with 4 μm thickness, on the charged glass slides. Antigen retrieval was completed with a heated 30% citrate buffer and 10% normal goat serum as the blocking solution. The immunohistochemistry staining was conducted with the antibody of Ki-67 (ab16667), with 1:200 dilution. The skin tissue sections were incubated with the antibody of Ki-67 (primary antibody) overnight, in the fridge (4 °C). On the second day, the tissue sections were washed with PBS and the 3′3-diaminobenzidine tetrahydrochloride (DAB) was used as the chromogen. Next, the tissue sections were incubated with the secondary antibody of goat anti-rabbit IgG H&L HRP-conjugate (AP307P) in 20% bovine serum albumin (BSA), with 1:100 dilution for 1 h, at room temperature, and counterstained with haematoxylin. The immunohistochemistry of Ki-67 was reported in the percentage of the immunoreactive nuclei using the formula of the number of immunoreactive nuclei (brown-stained nuclei) over the number of nuclei (a total of both unstained and stained nuclei) [39]. The quantification of Ki-67 was conducted with a minimum of five random areas of skin images under 40× magnification of the light microscope.

### 2.7. Statistical Analysis

Statistical Package for Social Sciences (SPSS) version 26 was used to analyse the quantitative data using the One-way analysis of variance (ANOVA). The comparison between the control and oral pterostilbene-treated groups was conducted using Dunnett’s post hoc test. All the quantitative data were presented as mean ± standard error mean (SEM). There is a significant difference between the groups if a *p*-value (probability value) of the statistical test is less than 0.05 (*p* < 0.05) and no significant difference if a *p*-value is more than 0.05 (>0.05).

## 3. Results

### 3.1. Inhibitory Effect of Orally Administered Pterostilbene on Specific Multistage Carcinogenesis of DMBA/TPA Induced Skin SCC Mouse Model

Macroscopic observations of the appearance of the tumours on the shaved dorsal skin, taken for the representative mouse from all groups at the end of the experimental animal period (week 24), are shown in Figure 3A. The topical application of 70% acetone did not cause any tumour development, and the shaved dorsal skin still maintained the normal gross appearance of a representative mouse from the Vehicle group, as shown in Figure 3A. In contrast, the dorsal skin of the representative mouse of the Cancer group, which was treated with DMBA/TPA and without oral pterostilbene, developed many tumours, as displayed in Figure 3A. Compared to the Cancer group, the macroscopic observations of the oral pterostilbene-treated groups, including the Initiation, Promotion and Continuous groups, showed significantly less tumour development on the dorsal skin, as displayed in Figure 3A. The graph in Figure 3B presents the percentage of mice that developed at least one tumour bigger than 1mm in diameter, from each group, from week 0 until week 24. The development of the tumour started at week 10 in the Cancer group. They were slightly delayed in first tumour development in the oral pterostilbene groups, as the first tumour was observed during week 11 for the Initiation group; week 12 for the Promotion group; and week 13 for the Continuous group. All the DMBA/TPA-treated groups achieved 100% of mice with tumours (100% incidence rate of tumour development) at week 24, before the mice were sacrificed. However, the oral pterostilbene treatment caused the delay in the Initiation, Promotion, and Continuous groups’ achieving 100% mice with tumours, as the Cancer group reached 100% mice with tumours much earlier, at week 12, marked by the red line in Figure 3B. The blue line of Figure 3B shows that the Promotion group achieved 100 % mice with tumours at week 17, which was five weeks delayed, compared to the Cancer group. The Initiation and Continuous oral pterostilbene-treated groups achieved 100 % mice with tumours six weeks later, compared to the Cancer control group, at week 18, marked by the green line in the graph of Figure 3B. The graph in Figure 3C shows the effect of oral pterostilbene on tumour multiplicity, or the average number of tumours per mouse, from week 0 until week 24. No tumour was observed in the Vehicle group from week 0 until the end of week 24. Pterostilbene exhibited a chemo-preventive effect against tumour development as the use of oral pterostilbene during Initiation (5.00 ± 0.86), Promotion (4.33 ± 0.71), and Continuous (4.17 ± 1.14) significantly reduced the average number of tumours per mouse compared to the Cancer (9.33 ± 1.38) group (*p* < 0.05) at the termination of the animal experimental period of week 24.

Figure 4A shows the effect of oral pterostilbene on tumour volume per mouse from week 0 until week 24. Oral pterostilbene, which was given at specific multistage carcinogenesis during the initiation, promotion, or continuous, significantly reduced tumour volume per mouse, as there are significant differences between the Initiation (47.02 ± 21.51 mm^3^), Promotion (13.23 ± 5.25 mm^3^) and Continuous (39.23 ± 25.94 mm^3^) groups compared to the Cancer control group (400.59 ± 89.79 mm^3^) (*p* < 0.05). The graph of Figure 4B shows the comparison number of tumours, based on the tumour size, which is categorised based on tumour diameter. Tumours were divided into three size categories: the small tumour, with 0 to less than 3 mm in diameter; medium tumours, with 3 to less than 5 mm in diameter; and large tumours, with 5 mm or more in diameter. The number of small tumours shows no significant difference between the three oral pterostilbene-treated groups of Initiation (4.33 ± 0.80), Promotion (4.17 ± 0.79), and Continuous (3.33 ± 0.76), compared to the Cancer (4.00 ± 1.41) control group(*p* > 0.05). However, oral pterostilbene inhibited tumour growth, as the oral pterostilbene-treated groups significantly reduced the number of medium and large size tumours. The number of medium size tumours in the Initiation (0.5 ± 0.5), Promotion (0.17 ± 0.17), and Continuous groups (0.67 ± 0.33) significantly reduced compared to the Cancer group (3.17 ± 0.40). Similarly, the number of large size tumours in the Initiation (0.17 ± 0.17) and Continuous groups (0.17 ± 0.17) were significantly reduced compared to the Cancer group (2.17 ± 0.31). The Promotion group had no tumour larger than 5 mm in diameter.

The graph in Figure 4C shows the effect of oral pterostilbene on body weight. The Cancer group showed a slower body-weight-gain trend from week 0 to week 24 compared to the Vehicle group, which had a much higher body-weight-gain trend. At the end of the experimental period, in week 24, the Cancer group had lower body weight when compared to the Vehicle control group, yet that difference is not statistically different, as the body weight of the Cancer group (32.13 ± 1.39 mg) was slightly lower than the Vehicle group (35.32 ± 1.18 mg) at week 24 (*p* > 0.05). The three oral pterostilbene-treated groups showed lower weight-gain trends than the Vehicle group throughout the experimental animal period. However, there were no significant differences between oral pterostilbene-treated during Initiation (33.95 ± 1.34 mg), Promotion (32.57 ± 1.73 mg), and Continuous (4.00 ± 2.47 mg) groups compared to the Vehicle group at week 24 (*p* > 0.05). The oral pterostilbene showed a slightly higher trend in body-weight gain compared to the Cancer group throughout the experimental period, from the beginning of week 0 until the end of the experimental animal period of week 24. However, the higher body weight in the three oral pterostilbene-treated groups was not statistically higher compared to the body weight of the Cancer control group at week 24 (*p* > 0.05).

### 3.2. Orally Administered Pterostilbene during Initiation, Promotion or Continuous Reduced the Thickness of the Epidermis Layer

Figure 4D shows the graph of epidermis layer-thickness for all the groups. The DMBA/TPA exposure without oral pterostilbene in the Cancer control group caused a significant increase in the epidermal thickness, as the Cancer (135.44 ± 4.20 µm) control group was significantly higher in epidermal thickness compared to the Vehicle (18.11 ± 0.42 µm) control group. The oral pterostilbene during the Initiation (81.67 ± 6.28 µm), Promotion (66.94 ± 2.69 µm), and Continuous groups (64.33 ± 6.85 µm) also significantly increased the epidermal thickness compared to the Vehicle control group (*p* < 0.05). However, the oral pterostilbene significantly reduced the epidermal-thickness as the Initiation, Promotion and Continuous groups showed a significant reduction in epidermal-thickness in comparison to the Cancer control group (*p* < 0.05).

### 3.3. Effect of Oral Administered Pterostilbene on Semi-Quantitative Histopathological Scoring Based on H&E Staining

Table 1 shows the mean histopathological score for each group based on the observation of H&E staining under the light microscope. The histopathological score for the Vehicle group was 0, as all the mice from the Vehicle control group showed no skin histology changes and maintained the typical skin histology characteristics of the mouse, as described in the next section. In contrast, the Cancer control group scored the highest histopathological score, with 2.87 ± 0.07, as most of the skin histology lesions were squamous cell carcinoma (SCC), with few papilloma. The oral pterostilbene treatment resulted in the reduction in the histopathological score as the Initiation, Promotion and Continuous groups histopathological scores were lower compared to the Cancer control group. The Initiation group scored 1.53 ± 0.18, with most of the skin lesions being papilloma and epidermal hyperplasia. The histopathological score for Promotion was 1.27 ± 0.18, and 1.15 ± 0.07 for Continuous, in which the majority of skin lesions were epidermal hyperplasia, with only a few papilloma lesions observed.

### 3.4. Effect of Orally Administered Pterostilbene during Initiation, Promotion or Continuous on Histopathological Observation against Carcinogenesis of Skin SCC Induced by DMBA/TPA

Figure 5 shows the histopathological observation of the skin histology, from the representative mouse of each group, based on the H&E staining images under the light microscope. The skin histology of the control Vehicle group showed the normal skin histology of the mouse with a clear separation of the epidermis, dermis, and subcutaneous layers, as shown in Figure 5A. The epidermis layer still maintains the normal appearance without any thickening, and the dermis layer consists of sebaceous glands located adjacent to the hair follicles. At a higher magnification of 40×, Figure 5B shows that the Vehicle control group that showed normal mouse skin histology, with a thin epidermis layer, uniformly arranged epithelial cells, and keratin over the epidermis layer. In contrast, the Cancer control group skin histology showed thickening, hyperkeratinisation, hyper-proliferation, and a corrugated epidermis layer. Moreover, the Cancer control group displayed the squamous cell carcinoma (SCC) histological features, including the formation of keratin pearls in Figure 5C, and abundant mitotic bodies were observed, as shown in Figure 5D. The Cancer group showed an invasion of the basement membrane as marked by arrows in Figure 5C, and exhibited highly pleomorphic cells, including the disorganisation of cells with varying shapes and sizes and also hyperchromatic nuclei, as shown in Figure 5D. The oral pterostilbene during initiation caused less keratinisation, and the papilloma with basement membrane was still intact, as shown by the skin histology of a representative mouse of the Initiation group in Figure 5E. The higher magnification (40×) of the Initiation group skin histology image in Figure 5F displays fewer pleomorphic cells and nuclei, compared to the Cancer control group. Similarly, the effect of oral pterostilbene during the Promotion also led to fewer alterations in skin histology, compared to the Cancer group, with less thickening in the epidermis layer and keratinisation with the basement membrane still intact, as shown in Figure 5G. The higher magnification (40×) of the skin histology of a representative mouse from the Promotion group, demonstrated in Figure 5H, shows fewer pleomorphic cells and nuclei, with more organised epithelial cells in the epidermis layer. Figure 5I shows the skin histology of the Continuous oral pterostilbene group with epidermal hyperplasia, demonstrating fewer changes in skin histology. The higher magnification (40×) of the Continuous group in Figure 5J shows epidermal hyperplasia, with many more epithelial cells. However, the epithelial cells showed no changes in morphology without nuclear and cellular atypia, as each cell maintained the size and shape of normal epithelial cells, which were almost similar to the epithelial cell of the Vehicle control group.

### 3.5. Effect of Orally Administered Pterostilbene during Initiation, Promotion and Continuous in Chemically DMBA/TPA Induced Skin SCC Mouse Model on the Cell Proliferation Marker of Ki-67

To evaluate the effect of oral pterostilbene on cell proliferation activity, we performed the immunohistochemistry of Ki-67, which acts as a cell proliferation marker, as shown in Figure 6. Figure 6A shows the representative image of the Vehicle group with no immunoreactive nucleus detected, and some images showed low expression of Ki-67 (not shown). In contrast, the Cancer group demonstrated numerous immunoreactive nuclei with brown staining, as shown in Figure 6B. Three oral pterostilbene-treated groups demonstrated a reduction in Ki-67 expression as less immunoreactive nuclei, as displayed in the immunohistochemistry microscope images in Figure 6C–E. The graph in Figure 6 shows the quantification of immunohistochemistry of Ki-67, presented in the percentage of positive nuclei over the entire number of nuclei in the epidermis layer. The Cancer control group (18.05 ± 0.38%) caused significant up-regulation in Ki-67 expression in comparison to the Vehicle (0.91 ± 1.98%) control group (*p* < 0.05). Three pterostilbene-treated groups caused a slight increase in Ki-67 expression in comparison to the Vehicle group. However, the slight increase in the Ki-67 expression, or the percentage of immunoreactive nuclei, in oral pterostilbene groups of Initiation (4.77 ± 0.92%), Promotion (3.56 ± 0.72%) and Continuous (2.80 ± 0.60%) is not statistically significant to the Vehicle control groups (*p* > 0.05). In addition, oral pterostilbene reduced the effect of DMBA/TPA-induced cellular hyperproliferation as the Initiation, Promotion and Continuous groups had lower expression of Ki-67 than the Cancer control group, with a statistically significant reduction in the percentage of immunoreactive nuclei Ki-67 (*p* < 0.05).

## 4. Discussion

The DMBA/TPA chemically induced skin SCC mouse model is a widely used cancer model that involves the mechanisms of multistage carcinogenesis, which are also common in the development of human cancers. Generally, multistage carcinogenesis consists of a few stages that start with initiation, promotion, progression, and, lastly, malignancy [40]. The DMBA/TPA induced non-melanoma skin cancer, specifically the squamous cell carcinoma (SCC) subtype, at the epidermis layer. The initiation stage involved DNA damage, as metabolised carcinogens will covalently bind to the DNA, and results in mutation. No morphological changes in skin histology occurred during the initiation stage. Next is the promotion stage, which includes the stimulation of cell proliferation that results in the hyperplastic epidermis. The promotion stage is followed by the progression stage, with the development of pre-malignant papilloma with more genetic changes. Later, in the progression stage, the papilloma will progress and be converted to SCC. Lastly, the conversion of SCC to invasive carcinoma involves the invasion of surrounding tissues and metastasis into other organs [34,41]. The potential application of this model is to study the effect of the dietary intervention or chemo-prevention on the specific stage of initiation and/or promotion, as the protocol DMBA/TPA-induced skin carcinogenesis mouse model technically distinguished the initiation and promotion stages [35,36]. During the initiation stage, the topical exposure of DMBA, the polyaromatic hydrocarbons, will result in DNA damage in which DMBA primarily targets the gene mutation of the H-ras oncogene at the 61^st^ codon. This mutation becomes the factor for the activation of the Ras protein and continues to stimulate cell growth via the persistence of signal transduction [42,43]. The promotion stage by the tumour promoter agent (TPA) of small molecule compound is very potent as an inducer for rapid activation of the protein kinase C (PKC) family [44]. PKC is a large group of proteins that include serine/threonine kinases, which are involved in broad cellular biological events that are related to carcinogenesis, including cell cycle, proliferation, and differentiation [45]. Even though DMBA is a very potent carcinogen that can act as both an initiating and a promoting agent, the two-week exposure of DMBA in this study did not cause the development of a tumour; this observation can be supported by a previous study, in which the authors reported that the exposure to DMBA alone, without a promoting agent, requires at least seven weeks to cause skin tumour development [46]. Unlike DMBA, the TPA can act only as a promoting agent; the TPA exposure alone, without initiation, did not cause tumour development, even after 30 weeks of TPA exposure [47]. In addition, TPA exposure alone has been proven to only cause tumour development in mice with activation-induced cytidine deaminase (AID), which induces the mutation of the H-ras gene. The mice which did not have DMBA exposure or AID expression, and were exposed to TPA alone, did not develop any tumours after 20 weeks [48]. Our results demonstrate that the TPA acted as a promoter agent on initiated cells in multistage carcinogenesis. Hence, in our study, we successfully induced the initiation and promotion in the DMBA/TPA-induced skin SCC mouse model, and distinguished the initiation and promotion stages of multistage carcinogenesis with two weeks of exposure to DMBA, followed by repeated exposure to TPA for 20 weeks, to expand the clonal expansion of initiated cells.

In our study, we revealed the effect of orally administered pterostilbene on specific multistage carcinogenesis during the initiation, promotion, and continuous stages in the DMBA/TPA-induced skin squamous cell carcinoma mouse model. Beyond our study, the potential of pterostilbene as chemoprevention against skin cancer has been reported, as the topical application of pterostilbene during the promotion and progression by TPA (after the initiation by DMBA) has been shown to exhibit a chemo-preventive effect, with the reduction in tumour multiplicity against carcinogenesis of skin SCC- induced chemically by DMBA/TPA [26]. Although there has been a study on the chemo-preventive effect of pterostilbene against skin cancer, induced by DMA/TPA, the effect of pterostilbene on specific multistage carcinogenesis during the initiation and promotion in the DMBA/TPA-induced skin carcinogenesis mouse model has yet to be determined. Hence, in our study, oral pterostilbene was given during the initiation, promotion or continuous stages in order to explore its anti-initiation and anti-promotion against skin carcinogenesis in the mouse model induced by DMBA/TPA. As an effective chemo-preventive agent, pterostilbene should be able to interrupt the early stage of carcinogenesis, especially during the initiation and promotion, as the progression involves more abnormalities in the genotype and phenotype of the lesion, and at this stage, the tumour grows rapidly and has an increased risk of invasion and metastasis. The chemo-preventive agent that targets the initiation and promotion of carcinogenesis is a promising intervention as the initiation stage of skin cancer involved no remarkable cellular and tissue morphology changes, and the promotion stage of skin cancer showed localised and slower tumour growth, with the potential of regression if the promoting agent is eliminated [49]. Moreover, our study is the first to investigate the chemo-preventive effect of orally administrated pterostilbene against the skin cancer mouse model induced by DMBA/TPA. Other than the high oral bioavailability of pterostilbene, the route of oral administration of pterostilbene was chosen as it is favourable in reaching the systemic circulation. Hence, this route of administration is more suitable for prophylaxis or chemo-preventive intervention, as the location of initiation in skin carcinogenesis cannot be predicted [50]. Moreover, the delivery of compounds via systemic circulation can also help to protect other internal organs, as skin SCC has the potential for metastasis, and metastatic skin SCC is lethal, as many studies have reported more than 70% of the mortality rate in skin SCC patients with metastasis [51]. The common organs that metastasis skin SCC tends to spread to are the lungs, liver, brain, lymph nodes, and bones [52,53]. In addition to this study, many in vivo studies have reported that the oral administration of pterostilbene inhibited the tumour growth of various types of cancer, including hepatocellular, endometrial, pancreatic, and breast carcinoma, in xenograft mouse models [54,55,56,57]. The reduction in epidermal layer-thickness in the oral pterostilbene groups within our study also showed that pterostilbene inhibited the skin SCC carcinogenesis, as this type of skin cancer originated from keratinocytes of the epidermis layer [58].

In addition, we also demonstrated that the administration of oral pterostilbene, during initiation, suppressed the carcinogenesis of skin SCC by reducing the number and volume of tumours and reduced the severity of histopathological changes. The anti-initiation effect of oral pterostilbene can result from the protective effect or repair mechanisms of pterostilbene in reducing DNA damage during the initiation stage by DMBA. Pterostilbene has been shown to induce cell senescence for DNA damage response, via p53-dependent mechanisms, in lung cancer cells [59]. Cell senescence is crucial for DNA damage response as it involves the cell cycle progression arrest in stopping the proliferation of cells, until the DNA damage is repaired, or the damage is removed [60]. Other than the induction of cell senescence, p53, also known as the guardian of the genome, is directly engaged in DNA repair processes such as base excision repair (BER) and nucleotide excision repair (NER) [61]. Moreover, pterostilbene has been shown to up-regulate the p53 protein expression, to exert a chemo-preventive effect against the carcinogenesis of lung squamous cell carcinoma and hepatocellular carcinoma in mouse models [62,63]**.** In addition to cell senescence and the DNA repair mechanisms of p53 as a tumour suppressor, p53 can also induce apoptosis cell death via receptor-mediated apoptosis and mitochondria-mediated apoptosis [64]. Thus, the anti-initiation by oral pterostilbene in our study could be a result of the DNA repair, cell senescence, and apoptosis mechanisms in p53-dependent pterostilbene. We also demonstrated the anti-promotion effect of oral pterostilbene during the promotion stage by TPA exposure. TPA has been proven to stimulate cell proliferation, acting as a promoter agent, in the overexpression of cell proliferation markers of PCNA and Ki-67 of mice skin with exposure to TPA [65]. The effect of oral pterostilbene as an anti-promotion in our study can be supported by the previous study, as dietary intake of pterostilbene significantly reduced the expression of PCNA in azoxymethane-induced colon cancer in rats [66].

Moreover, TPA has been proven to promote tumour development, via the elicitation of the inflammatory response, as TPA up-regulated the production of enzymes, which can trigger inflammation, such as inducible nitric oxide synthase (iNOS) and cyclooxygenase-2 (COX-2) [67]. The anti-inflammatory property of pterostilbene can contribute to its anti-promotion effect, as topical pterostilbene has been shown to reduce the protein expression of iNOS and COX-2 in this chemically induced skin SCC mouse model by DMBA/TPA. In the same study, the topical application of pterostilbene also down-regulated the activation of NF-κB by TPA, as pterostilbene reduced the phosphorylation of NF-κB and its inhibitor of Iκbα to inhibit the inflammatory response [26]. Other than skin SCC, pterostilbene has been proven to exert a chemo-preventive effect against another type of SCC via its anti-inflammatory effect. Pterostilbene inhibits the NF-κB inflammatory signaling pathway via the reduction in phosphorylation of Iκbα in human oral SCC cell lines [68]. The molecular mechanisms, via the modulation of the NF-κB signaling pathway of pterostilbene, as a chemo-preventive agent is supported by the vital role of the NF-κB signaling pathway in the development of cancer, including skin squamous cell carcinoma. The NF-κB has been reported to play a vital role in the development of skin cancer in the DMBA/TPA-induced skin carcinogenesis mouse model, as the mice which had a depletion of NF-κB, due to the deletion of the NF-κB gene, showed a significantly reduced tumour incidence compared to the normal mice. In other words, NF-κB is essential to promote skin carcinogenesis in the mouse model, as deficiency of NF-κB suppressed the carcinogenesis of skin squamous cell carcinoma. In addition, the deficiency of NF-κB prevents cell death after DNA damage, as well as the inhibition of inflammation via the downregulation of cytokines and chemokines genes expression, such as tumour necrosis factor (TNF) and C-X-C motif chemokine ligand 1 (Cxcl1) genes [69]. Hence, the anti-inflammatory property of pterostilbene may be responsible for its anti-promotion effect in inhibiting the carcinogenesis of skin SCC in this study. Immunohistochemistry of Ki-67 is one of the methods used to evaluate the cell proliferation index on mouse skin [70]. Ki-67 is a non-histone nuclear protein and a reliable cell proliferation marker, as Ki-67 expression occurs during the active phases of the cell cycle (G_1_, S, G_2_ and M) and goes through rapid degradation until it becomes absent during the gap phase of G_0,_ where the cells are not actively dividing in the early phase of G_1_ [71,72]. The DMBA/TPA-induced skin carcinogenesis mouse model has previously been proven to promote cell proliferation as overexpression of Ki-67 has been observed in the previous study using this skin SCC mouse model [73]. Similar to our findings, cancer control, treated with DMBA/TPA, caused a significant increase in the percentage of immunoreactive nuclei of Ki-67. Moreover, our results showed that oral pterostilbene, which was given during the initiation and/or promotion in DMBA/TPA-induced multistage skin SCC carcinogenesis, significantly reduced cell proliferation via the downregulation of Ki-67. This finding is supported by another in vivo study, in which the diet containing pterostilbene was shown to slow down cell proliferation activity via the significant reduction in the expression of Ki-67 in a prostate cancer transgenic mouse model [74]. Moreover, the potential of pterostilbene as a chemo-preventive agent can be supported by findings of previous studies on its natural analogues against the development of squamous cell carcinoma. For example, resveratrol, which is a widely studied natural analogue of pterostilbene, has been shown to significantly reduce tumour occurrence in DMBA/TPA skin cancer mouse models via the induction of cell cycle arrest and apoptosis [75]. In addition, the 3′-hydroxypterostilbene, also a natural analogue of pterostilbene, has been reported to inhibit the development of skin squamous cell carcinoma in mouse models induced by DMBA/TPA. Treatment with 3′-hydroxypterostilbene during the initiation or continuously stages caused a significant reduction in the number of tumours and the downregulation of protein expression of inflammation and cell proliferation markers [76].

## 5. Conclusions

Overall, the current results of our study demonstrate that oral pterostilbene is a promising chemo-preventive agent in interrupting the development of skin squamous cell carcinoma, as pterostilbene exerts anti-initiation and anti-promotion effects against carcinogenesis of skin SCC in this chemically induced mouse cancer model of DMBA/TPA. However, further investigation at molecular levels of the potential mechanism of orally administered pterostilbene, such as p53 and anti-inflammatory, via the regulation of NF-κB signaling pathways in the prevention of skin SCC development, is needed.

## Figures and Tables

**Figure 1 biomedicines-10-02743-f001:**
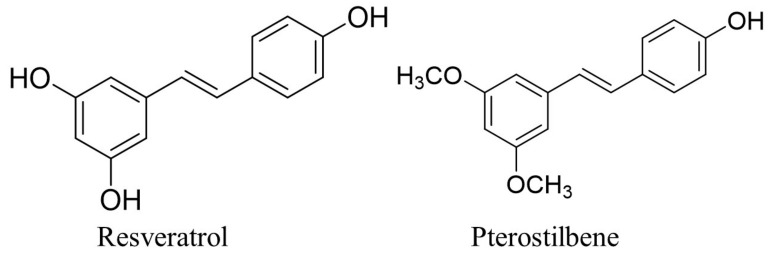
The chemical structure of resveratrol and pterostilbene [29].

**Figure 2 biomedicines-10-02743-f002:**
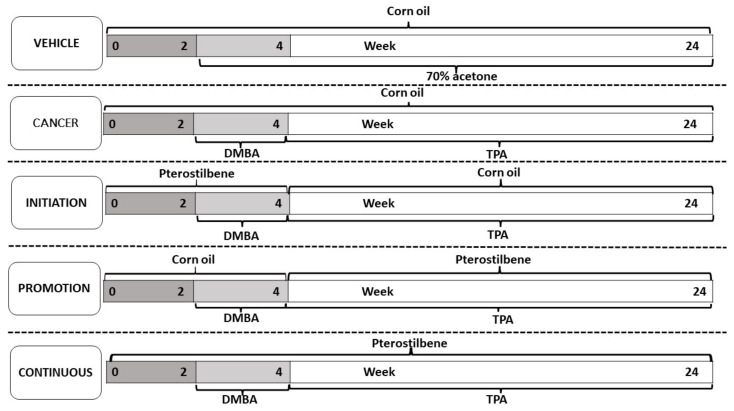
The experimental design of the study for each group was to investigate the chemo-preventive effect of oral pterostilbene during the initiation, promotion, or continuous in DMBA/TPA induced skin carcinogenesis mouse model.

**Figure 3 biomedicines-10-02743-f003:**
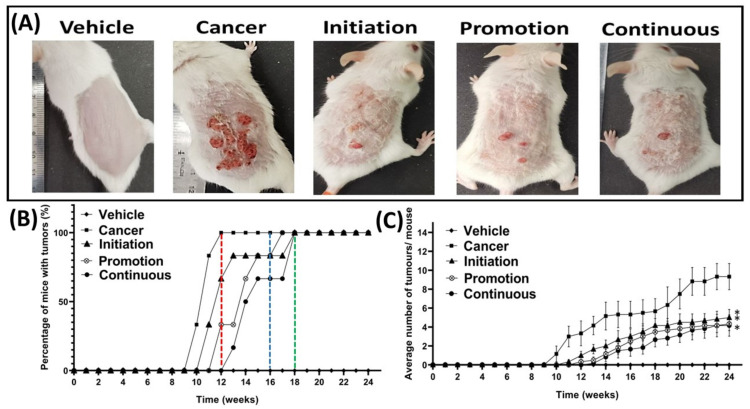
Effect of oral pterostilbene during the initiation, promotion, or continuous in a mouse model of skin SCC induced chemically by DMBA/TPA. (**A**) The macroscopic observation of the dorsal skin of representative mice from the Vehicle, cancer, Initiation, Promotion, and Continuous groups at week 24. (**B**) the percentage of mice with tumours with a red line marked at week 12 where the Cancer control group achieved a 100% incidence rate, a blue line marked the Promotion group achieved a 100% incidence rate at week 17, and a green line marked the Initiation, and Continuous groups achieved 100% at week 18. (**C**) The average number of tumours per mouse (mean ± SEM). * there is a statistically significance difference to the Cancer control group (*p* < 0.05) at week 24.

**Figure 4 biomedicines-10-02743-f004:**
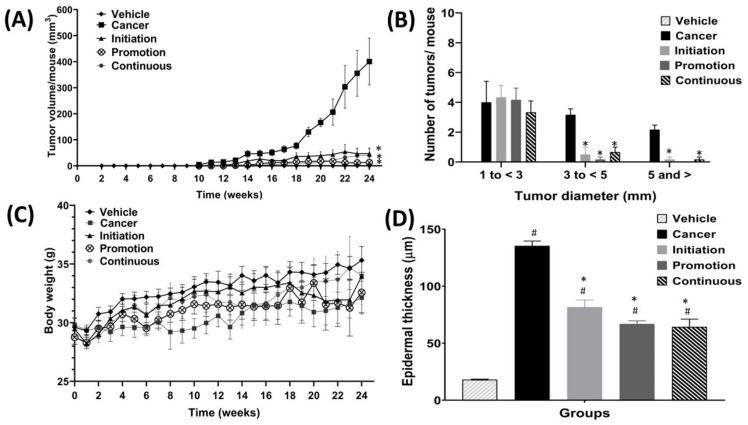
The effect of oral pterostilbene during the initiation, promotion, and continuous on tumour volume, tumour diameter categories, body weight, and epidermal thickness. (**A**) The tumour volume per mouse for each group from week 0 to week 24. (**B**) The tumour diameter categories (small, medium, and large) at week 24. (**C**) The body weight of all groups from week 0 to week 24. (**D**) The epidermal thickness of all groups. * there is a statistically significant difference compared to the Cancer group (*p* < 0.05) at week 24. ^#^ there is a statistically significant difference to the Vehicle control group (*p* < 0.05) at week 24.

**Figure 5 biomedicines-10-02743-f005:**
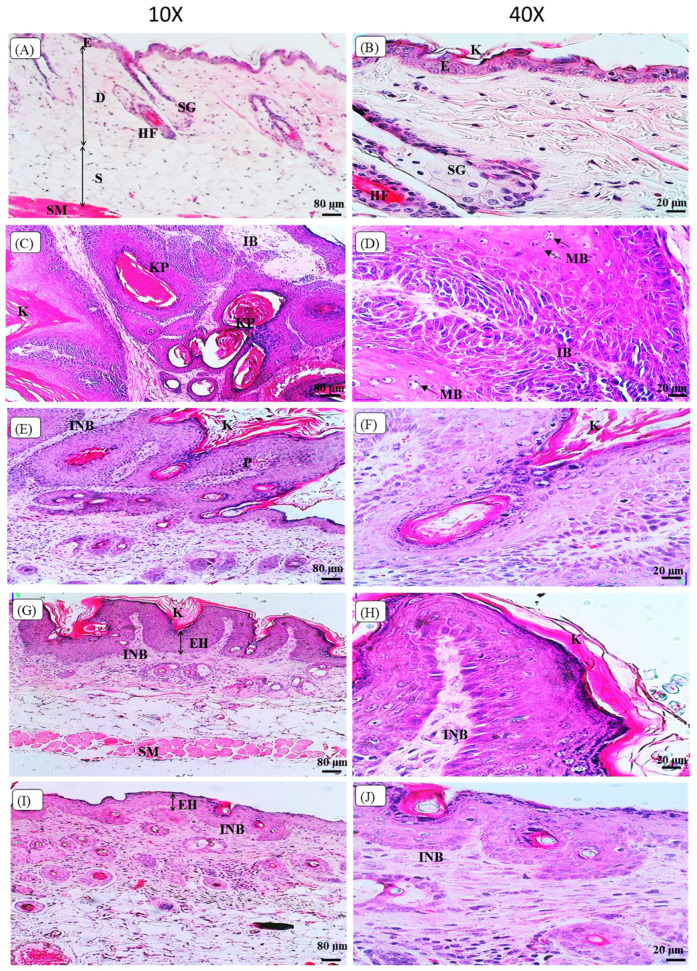
The images of histopathological observation of Hematoxylin and Eosin (H&E) staining under the light microscope. (**A**,**B**) are the skin histology of the Vehicle group with 10× and 40× magnification, respectively. (**C**,**D**) are the skin histology of the Cancer group with 10× and 40× magnification, respectively. (**E**,**F**) are the skin histology of the Initiation group with 10× and 40× magnification, respectively. (**G**,**H**) are the skin histology of the Promotion group with 10× and 40× magnification, respectively. (**I**,**J**) are the skin histology of the Continuous group with 10× and 40× magnification, respectively. E: epidermis layer; D: dermis layer; S: subcutaneous layer; SM: smooth muscle; HF: hair follicle; SG: sebaceous gland; K: keratin; KP: keratin pearl, IB: invasion of the basement membrane, MB: mitotic bodies, INB: intact basement membrane; P: papilloma; EH: epidermal hyperplasia.

**Figure 6 biomedicines-10-02743-f006:**
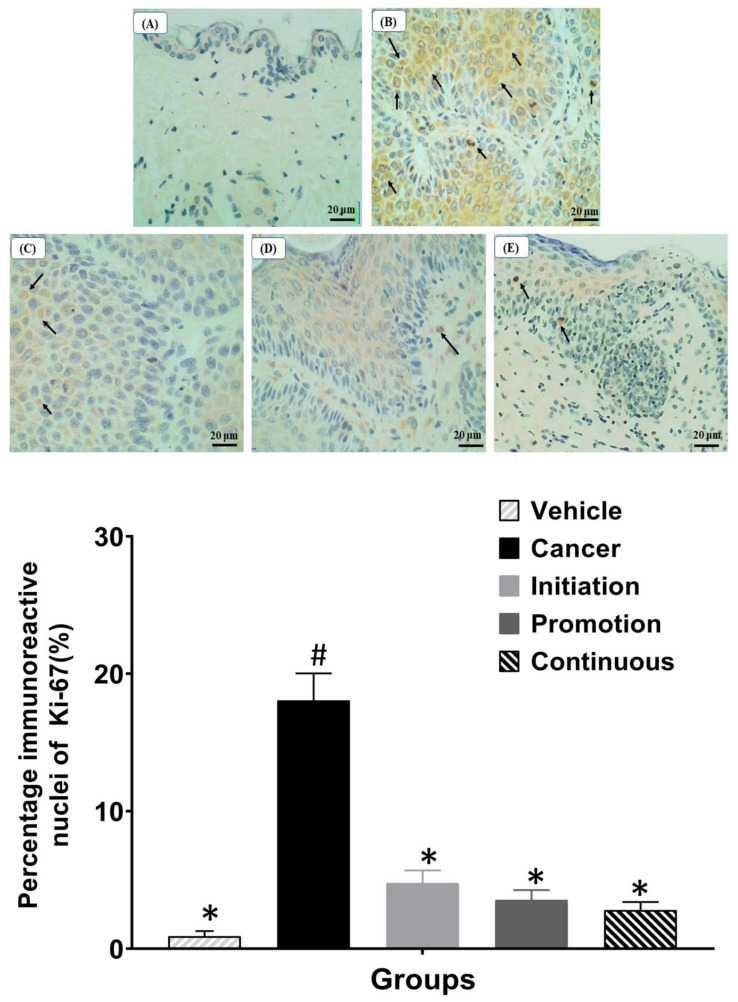
The images of immunohistochemistry of Ki-67 staining under the light microscope of 40× magnification. (**A**): Vehicle, (**B**): Cancer, (**C**): Initiation, (**D**): Promotion, and (**E**): Continuous. The graph shows the percentage of immunoreactive nuclei of Ki-67 for each group. Arrows show the immunoreactive nuclei of Ki-67 with brown staining. * there is a statistically significant difference to the Cancer control group (*p* < 0.05) at week 24. ^#^ there is a statistically significant difference to the Vehicle group (*p* < 0.05) at week 24.

**Table 1 biomedicines-10-02743-t001:** Histopathological score based on hematoxylin and eosin (H&E) staining.

Groups	Histopathological Score	Skin Lesions
VEHICLE	0	Normal
CANCER	2.87 ± 0.07	Squamous cell carcinoma
INITIATION	1.53 ± 0.15	Papilloma
PROMOTION	1.27 ± 0.18	Hyperplasia
CONTINUOUS	1.15 ± 0.07	Hyperplasia

## Data Availability

Not applicable.

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
