# Peer review of "Chemopreventive Effects of Oral Pterostilbene in Multistage Carcinogenesis of Skin Squamous Cell Carcinoma Mouse Model Induced by DMBA/TPA"

_biomedicines, 2022, doi:10.3390/biomedicines10112743_

Round 1
Reviewer 1 Report
This article deals with the usefulness of pterostilbene in dealing with squamous cell carcinomas. The manuscript is well written, the English language is good while the figures are clear.
There is one part of this document that could be improved namely the paragraph consisting of lines 45 – 59 in the following ways. (1) The structures of pterostilbene and resveratrol should presented. (2) The question of differences in the lipophilicity of pterostilbene and resveratrol is mentioned and some evidence is required. One could point out for example that the Hansch pi values of the methoxy and hydroxy groups are -0.02 and -0.67, respectively, to support this assertion. (3) Detoxification of a free hydroxy group would be expected to be facile whereas the methoxy substituents would need to undergo O—demethylation. In other words, the methyl groups protects the hydroxy substituent from rapid metabolism.
Another area which could be included in this study is drawing the attention of the reader to any analogs of pterostilbene that have been evaluated for growth-inhibiting properties of squamous cell carcinomas.
In summary this manuscript is recommended for publication in Biomedicines although the paragraph of lines 45 – 59 could be strengthened.(If the authors need a reference to Hansch pi values they will find the following monograph useful : C. Hansch and A.J. Leo, Substituent constants for correlation analysis in chemistry and biology, John Wiley and Sons, New York, 1979, p.49).
In summary this manuscript is recommended for publication in Biomedicines although the paragraph of lines 45 – 59 could be strengthened.(If the authors need a reference to Hansch pi values they will find the following monograph useful : C. Hansch and A.J. Leo, Substituent constants for correlation analysis in chemistry and biology, John Wiley and Sons, New York, 1979, p.49).
Author Response
Greetings Dr/Prof,
Thank you very much for your comments and suggestions on this manuscript. We have done the amendations based on your comments and suggestions. Please see the attachment for our response to the comments.
Thank you

Reviewer 2 Report
The manuscript titled "Chemopreventive effects of oral pterostilbene in multistage carcinogenesis of skin squamous cell carcinoma mouse model induced by DMBA/TPA" describes the results of an in vivo mouse study of oral pterostilbene administered during skin carcinogenesis.
The study is not particularly novel as pterostilbene chemopreventive activity has been previously extensively reported in many disease models, including in skin cancer. Given that the molecular mechanisms of pterostilbene-associated chemoprevention are fairly well elucidated, at a minimum the outcome of those markers (which the authors described in the discussion, based on previous literature) should be included in the manuscript.
Author Response
Greetings Dr/Prof,
Thank you very much for your comments and suggestions on our manuscript. We have done the amendations of this manuscript based on your comments and suggestions. Please see the attachment for our response to the comments.
Thank you

Round 2
